# Peptidergic Systems and Neuroblastoma

**DOI:** 10.3390/ijms26083464

**Published:** 2025-04-08

**Authors:** Manuel Lisardo Sánchez, Rafael Coveñas

**Affiliations:** 1Laboratory of Neuroanatomy of the Peptidergic Systems, Institute of Neurosciences of Castilla and León (INCYL), University of Salamanca, 37007 Salamanca, Spain; lisardosanchez8@gmail.com; 2Group GIR USAL: BMD (Bases Moleculares del Desarrollo), University of Salamanca, 37007 Salamanca, Spain

**Keywords:** oncogenic peptides, anticancer peptides, peptide receptor antagonists, gastrin-releasing peptide, neuropeptide Y, RC-3095, aprepitant, BIIE0246

## Abstract

The peptidergic systems are involved in neuroblastoma. Peptides (angiotensin II, neuropeptide Y, neurotensin, substance P) act as oncogenic agents in neuroblastoma, whereas others (adrenomedullin, corticotropin-releasing factor, urocortin, orexin) exert anticancer effects against neuroblastoma. This plethora of peptidergic systems show the functional complexity of the mechanisms regulated by peptides in neuroblastoma. Peptide receptor antagonists act as antineuroblastoma agents since these compounds counteracted neuroblastoma cell growth and migration and the angiogenesis promoted by oncogenic peptides. Other therapeutic approaches (signaling pathway inhibitors, focal adhesion kinase inhibitors, peptide receptor knockdown, acetic acid analogs) that also counteract the beneficial effects mediated by the oncogenic peptides in neuroblastoma are discussed, and future research lines to be developed in neuroblastoma (interactions between oncogenic and anticancer peptides, combination therapy using peptide receptor antagonists and chemotherapy/radiotherapy) are also suggested. Although the data regarding the involvement of the peptidergic systems in neuroblastoma are, in many cases, fragmentary or very scarce for a particular peptidergic system, taken together, they are quite promising with respect to potentiating and developing this research line with the aim of developing new therapeutic strategies to treat neuroblastoma in the future. Peptidergic systems are potential and promising targets for the diagnosis and treatment of neuroblastoma.

## 1. Introduction

Neuroblastoma (NB), a disease with survival rates below 50%, is the most common extracranial solid tumor appearing in the childhood. NB results from an uncommon development of neural crest progenitor cells of the sympathetic nervous system. Chemoradiation, autologous stem cell rescue, and surgical resection are current therapeutic tools to fight this disease [1]. Other anti-NB strategies are the use of phosphoinositide 3-kinase, fibroblast growth factor receptor, or cyclin-dependent kinase 3/6 inhibitors [2]; immunotherapy [3,4]; NB-extracellular vesicles targeting [5]; natural killer cells [6,7]; and retinoic acid therapy [8]. Moreover, a new stemness-related predictive model for NB by means of bioinformatics/machine learning examination has been suggested [9], and a recent review on precision medicine applications in NB has recently been published [10]. There is therefore a plethora of anti-NB strategies that are being developed. In addition, another promising line of research is the participation of the peptidergic systems in the progress of tumors because not only do peptides exert antitumor effects, they also favor tumor progression; that is, peptides act as anticancer or oncogenic factors [11]. Peptides contain less than 50 amino acids and can be synthesized and released from nerve cells, as well as from endothelial and epithelial cells, cells of the immune system (e.g., T cells, eosinophils, macrophages), and tumor cells [11,12]. This means that there are numerous sources where peptides are synthesized and released, and they are involved in numerous physiological (e.g., cardiovascular and respiratory mechanisms, learning, memory, bone metabolism, thermoregulation, food intake, immunoregulation, sleep, glucoregulation) and pathological (e.g., asthma, urinary tract dysfunction, nausea, vomiting, inflammation, pain, neurotrauma, stroke, Alzheimer’s disease, depression, stress, anxiety) actions. Tumor cells produce and release peptides that exert autocrine, paracrine, and endocrine actions as peptides are released into the bloodstream [11,12]. Moreover, tumor cell activity is regulated by peptides released from nerve endings, endothelial cells, and immune cells located in the tumor microenvironment [11,12]. Peptide receptor antagonists are potential anti-NB agents because they stimulate apoptosis in tumor cells and block the migration/invasion of these cells, as well as inhibiting angiogenesis [12]. Moreover, peptide receptor antagonists combined with radiotherapy/chemotherapy decreased the side-effects mediated by cytostatics and also exerted an anticancer synergic action [13,14]. Accordingly, this review is focused on the anticancer and oncogenic peptides involved in NB and on the therapeutic anti-NB strategies resulting from the actions mediated by the peptidergic systems. Finally, future research lines will be suggested.

## 2. Neuroblastoma and Peptidergic Systems

### 2.1. Oncogenic Peptides

#### 2.1.1. Amylin

Human amylin promoted the release of factors (vascular endothelial growth factor, HspB5) by endothelial cells, favoring the proliferation and survival of human SH-SY5Y NB cells [15].

#### 2.1.2. Angiotensin II

The expression of angiotensin II receptor 1 (ATR1) in human NB cell lines (CHP-404, CHP-134B, SMS-K-CNR) has been reported, but in others, this expression was not observed (CHP-100, GOTO, SK-N-AS) [16]. Moreover, in the same study, it was demonstrated that the vasoactive intestinal polypeptide increased the density of ATR1 in NB cells and that angiotensin II did not favor the synthesis of DNA in these cells [16]. However, angiotensin II acted as a growth factor in human SH-SY5Y NB cells and insulin (at high concentration), and insulin-like growth factor-1 (at low concentration) increased the proliferative effects mediated by angiotensin II on these cells [17]. The effect mediated by insulin was blocked with angiotensin II receptor antagonists such as DuP753 (ATR1 blocker) or PD123177 (ATR2 blocker) [17]. Moreover, in the human SK-N-MC NB cell line, the expression of the angiotensin 4 receptor has been described [18]. This receptor, after binding to angiotensin IV (an angiotensin II fragment), promoted the synthesis of DNA in SK-N-MC NB cells. This was also induced by the peptide LVV-hemorphin 7 [18].

Furthermore, angiotensin II promoted the differentiation of human SH-SY5Y NB cells by increasing the levels of reactive oxygen species and microtubule-associated protein (MAP) 2 without affecting viability. This increase was also mediated by the activation of NADPH oxidase [19]. SH-SY5Y NB cells, treated with angiotensin II, decreased in size, and when these cells were treated with the ATR2 inhibitor PD123319, a reduction in the level of MAP2 was found. This means that angiotensin II regulated the differentiation of NB cells by increasing MAP2 level via AT2R [19]. Moreover, angiotensin II, as well as the ATR2 agonist CGP42112A, favored neurite outgrowth/beta III-tubulin expression in SH-SY5Y NB cells [20]. This means that these mechanisms were mediated by ATR2, and in fact, the ATR2 antagonist PD123319 counteracted the effects mediated by angiotensin II or CGP42112A. Thus, the ATR2 signaling pathway plays an important role in neuronal differentiation. Neurite growth mediated by CGP42112A required the activation of c-Src, sphingosine kinase, and mitogen-activated protein kinase but not phosphatidylinositol 3-kinase stimulation [20]. The data reported in this study also suggested a transactivation of the nerve growth factor receptor tyrosine kinase A, since its inhibition decreased neurite outgrowth mediated by angiotensin II or CGP42112A.

#### 2.1.3. Bradykinin

The effects mediated by bradykinin have been studied in human NB cell lines (CHP-100, CHP-134, IMR-32, SH-SY5Y) [21]. Bradykinin increased metalloproteinase activity and chemotactic response of NB cells, favored cell adhesion, and upregulated the expression of the vascular endothelial growth factor [21]. Moreover, bradykinin augmented the expression of the truncated purinergic P2X7B receptor (compared to the P2X7A full-length form expression) and promoted the proliferation of tumor cells, which was counteracted with purinergic P2X7 receptor antagonists. Mice transplanted with bradykinin-treated NB cells showed higher metastasis rates than animals administered with non-treated cells, and animals that received a purinergic P2X7 receptor antagonist (Brilliant blue G) did not show dissemination of NB cells to liver and bone marrow [21]. The data show that bradykinin and purines play a crucial role in NB dissemination, favoring metastasis.

Bradykinin also favored neurite outgrowth via myristoylated alanine-rich C kinase substrate phosphorylation involving the protein kinase C-dependent ROCK/RhoA pathway and protein phosphatase 2A in SH-SY5Y NB cells [22]. Bradykinin-potentiating peptides promoted the actions mediated by bradykinin because these peptides blocked angiotensin-converting enzyme activity. One of these peptides, named 10c (a synthetic bradykinin-potentiating peptide), protected SH-SY5Y NB cells against H_2_O_2_-induced oxidative stress [23]. Thus, 10c decreased the cell death promoted by H_2_O_2_, lipid peroxidation, reactive oxygen species generation, NF-κB expression, nitrate level, and inducible nitric oxide synthase expression, and it also protected mitochondrial membranes against oxidation [23].

#### 2.1.4. Gastrin-Releasing Peptide/Bombesin

Gastrin-releasing peptide (GRP) acts as an autocrine growth factor in human NB, and the peptide is synthetized by NB cells [24,25]. GRP activated the phosphatidylinositol 3-kinase/Akt survival signaling pathway favoring DNA synthesis and cell cycle progression in NB cells; in fact, the silencing of the GRP receptor promoted cell cycle arrest [26,27]. GRP receptor was overexpressed in NB [1]. This overexpression downregulated phosphatase and tensin homologue (PTEN) expression, favoring NB cell growth [27]. The expression of both GRP and GRP receptor protein was reported in human NB tissues, and, importantly, an augmented receptor expression was observed in more aggressive and undifferentiated tumors in comparison with that reported in benign tumors [25]. Moreover, GRP receptor mRNA was found in human NB cell lines, and the treatment with GRP provoked the mobilization of intracellular calcium in SK-N-SH/LAN-1 NB cell lines and the growth of NB cells (SH-SY5Y, SK-N-SH, LAN-1, IMR-32) [25]. The latter effect was blocked when SK-N-SH NB cells were treated with anti-GRP antibodies. This means that immunotherapy could be a potential a promising therapeutic strategy against NB. In this sense, another study has reported the use of anti-GRP receptor monoclonal antibodies to treat NB; in this study, LAN-1, SK-N-DZ, SHEP, SH-SY5Y, SK-N-AS, and BE (2)-C NB cell lines were used [1]. Monoclonal antibodies inhibited the activation of GRP receptors by GRP and its downstream phosphatidylinositol 3-kinase/Akt signaling pathway. This blocked the proliferation of NB cells and promoted antibody-dependent cellular cytotoxicity in NB cells. Moreover, anti-GRP receptor monoclonal antibodies favored the release of interferon γ/cytotoxic granzyme B by natural killer cells and decreased the mitotic tumor cells in vivo [1].

GRP increased the migration of NB cells and matrix metalloproteinase-2 expression, whereas it decreased the level of TIMP metallopeptidase inhibitor-1 in human NB cells. The study in question was performed in IMR-32/LAN-1/SK-N-SH NB cells [28]. GRP receptor overexpression upregulated integrin α2/α3/β1 protein/mRNA expressions and favored SK-N-SH NB cell migration, whereas the silencing of integrin β1 blocked tumor cell migration [28]. Akt2 controls the metastatic potential of human NB cells (SK-N-BE (2), BE (2)-M17, BE (2)-C) [29]. The activation of the phosphatidylinositol 3-kinase/Akt signaling pathway has been correlated with a poor prognosis in patients suffering from NB [29]. This pathway plays a crucial role in NB oncogenic transformation, which is promoted by the GRP/GRP receptor system and requires the N-myc oncogene [29]. The expression of the latter oncogene was regulated by Akt2 in NB cells, and Akt2 attenuation favored anchorage-independent cell growth, decreased cell proliferation, and reduced the release of the angiogenic vascular endothelial growth factor [29]. Moreover, Akt2 silencing blocked the migration/invasion of NB cells, and Akt2 silencing NB cell xenografts resulted in fewer metastases in experimental animals compared to controls. The data suggest that the GRP/GRP receptor/Akt2 axis is involved in NB progression. Focal adhesion kinase (FAK), a downstream target of the GRP receptor signal, was involved in NB metastasis and tumorigenesis [30]. Its expression and that of GRP receptors were correlated in human NB, whereas in SK-N-SH NB cells, the overexpression of these receptors augmented integrin (α3/β1)/FAK expressions and cell migration [30]. GRP receptor silencing reduced Mycn/FAK levels in BE (2)-C NB cells; FAK overexpression favored cell growth in SK-N-SH NB cells; and FAK blockers (Y15) inhibited the NB cell growth and metastasis promoted by GRP [30]. Dichloroacetate (an acetic acid analog) reduced the proliferation of SK-N-AS/BE (2)-C NB cells and promoted cell death, and treatment with dichloroacetate and GRP receptor silencing synergistically blocked the proliferation of BE (2)-C NB cells [31]. RC-3095 reduced GRP release from NB cells and tubule formation by endothelial cells and increased the expression of pro-autophagic proteins in NB cell lines (BE (2)-M17, BE (2)-C) [32]. Moreover, the inhibition of GRP receptors increased the autophagy-mediated degradation of GRP in NB cells. Autophagy inhibited angiogenesis through GRP degradation in NB cells, and this means that autophagy activation is a potential and promising antivascular strategy in NB [32]. GRP receptor antagonists/GRP receptor knockdown blocked the growth of NB cells; however, depending on the administered dose of RC-3095, this GRP antagonist blocked (low concentration) or promoted (high concentration) the proliferation of murine neuro2a NB cells [33]. The proliferative effect mediated by RC-3095 was counteracted with sodium butyrate (a histone deacetylase inhibitor). The data suggest that GRP receptors could interact with epigenetic mechanisms regulating the growth of murine NB cells.

Changes in cell morphology, reduced cell proliferation and size, DNA synthesis inhibition, Akt downregulation, PTEN expression upregulation (inhibitor of the phosphatidylinositol 3-kinase/Akt signaling pathway; poorly differentiated NB shows reduced PTEN protein concentration), and anchorage-independent growth suppression was observed in GRP receptor knockdown BE (2)-C NB cells [34]. Moreover, GRP receptor deficiency reduced liver metastasis and deferred tumor growth in vivo [34]. GRP receptors control the metabolism of glucose by controlling hypoxia-inducible factor-1 alpha, pyruvate dehydrogenase phosphatase 2, and pyruvate dehydrogenase kinase 4; in addition, the latter controls glucose metabolism through the regulation of the hypoxia-inducible factor-1 alpha [31]. The silencing of the GRP receptor blocked key regulators of the aerobic glycolysis mechanisms in human NB cells (SK-N-AS, BE (2)-C) [31]. GRP receptor silencing reduced the expression of hypoxia-inducible factor-1 alpha and pyruvate dehydrogenase kinase 4 mRNA expression (both involved in aerobic glycolysis) and augmented the pyruvate dehydrogenase phosphatase 2 mRNA expression (involved in glucose oxidation) [31]. This is important since tumor cells, under normoxic conditions, use the aerobic glycolysis for energy production instead of glucose oxidation [31]. GRP silencing also regulated the cell signaling pathways involved in invasion and metastasis since NB cell migration (BE (2)-C, SH-SY5Y) in vitro and metastasis in vivo were blocked [24]. Furthermore, GRP silencing inhibited NB cell-mediated angiogenesis and reduced anchorage-independent growth and the mRNA levels of the oncogenes involved in NB progression. The study also reported that the PTEN/Akt signaling pathway is a key mediator of the tumorigenic properties of GRP in NB cells [24]. Cell migration mediated by GRP and angiogenesis decreased when PTEN was overexpressed, and Akt activation was positively correlated with NB progression. GRP, via GRP receptors, favored p21 (located in the nucleus)/p27 (cytoplasm) expressions in BE (2)-C NB cells, and the silencing of the GRP/GRP receptor signal augmented tumor suppressor PTEN expression/accumulation in the cytoplasm of NB cells [26]. The data suggest that cyclin-dependent kinase inhibitors (p21, p27) and tumor suppressors (PTEN) are involved in NB tumorigenesis. Finally, the overexpression of the transcription factor Ets 1 has been related to an upregulation of the GRP receptor promoter activity [35]. This factor is involved in the transcription of the GRP receptor in NB. This finding could serve to downregulate its expression; hence, the mitogenic action mediated by GRP on NB cells could be counteracted.

Bombesin and phorbol myristate acetate (a protein kinase C agonist) increased the release of vascular endothelial growth factor from BE (2)-C NB cells [36]. This effect was synergistic when GRP was administered. Thus, protein kinase C controlled the bombesin-mediated secretion of the vascular endothelial growth factor in NB cells, and when the secretion of this factor was inhibited, a reduced NB cell proliferation-induced by GRP was reported [36]. Bombesin mediated NB cell growth (BE (2)-C, SK-N-SH) and promoted the expression of angiogenic markers (vascular endothelial growth factor, platelet endothelial cell adhesion molecule (PECAM) 1), favoring angiogenesis [37]. The GRP receptor antagonists RC-3095 counteracted in vivo both the angiogenesis and tumor growth promoted by bombesin [37]. Bombesin increased Akt phosphorylation in NB cells (SK-H-SH, LAN-1, BE (2)-C), and this was counteracted when cells were treated with the GRP receptor antagonist H2756 [27]. The latter effect was also observed when NB cells were treated with LY-294002, a phosphatidylinositol 3-kinase signaling pathway inhibitor [27]. The data show that this pathway is important in NB cell growth mediated by bombesin.

#### 2.1.5. Neuropeptide Y

A recent review dedicated on the involvement of neuropeptide Y in cancer development, including neuroblastoma, has been published [38]. Human NB cell lines (NB39, NB45, NB52, NB726) have been heterotransplanted into athymic nude mice: NB39 are undifferentiated cells; NB45 are poorly differentiated cells showing undifferentiated components; NB52 are poorly differentiated cells; and NB726 are differentiated cells [39]. In this study, the four mentioned NB cells expressed neuropeptide Y, NB45/NB52/NB726 displayed galanin, NB45/NB726 expressed calcitonin gene-related peptide, and NB726 showed vasoactive intestinal peptide and enkephalins (methionine- and leucine-enkephalin) [39]. Moreover, the average doubling time of tumors was two (NB39), ten (NB45), twenty-two (NB52), and forty-five (NB726) days [39]. The data show that human NB cells have different biological features. The neuropeptide Y receptor 2 is expressed in NB cells [40], and SK-N-MC NB cells expressed neuropeptide Y receptors 1, 4, and 5 [41]. The latter receptor was involved in cell proliferation [42].

Released neuropeptide Y from NB cells exerted via the neuropeptide Y receptor 2, an autocrine action favoring the proliferation of these cells and angiogenesis [40,43]. This proliferative action was increased when NB cells also expressed the neuropeptide Y receptor 5 [42,44], and the synthesis/release of neuropeptide Y from NB cells were mediated by protein kinase C-coupled M3 muscarinic receptors [45,46]. NPY increased SH-SY5Y NB cell survival and counteracted the toxic effect induced by β-amyloid on these cells [47] and also protected these cells against glutamate toxicity and endoplasmic reticulum stress. Neuropeptide Y exerted anti-apoptotic mechanisms, increasing the viability/survival of NB cells [48]. Neuropeptide Y, via the neuropeptide Y receptor 5/RhoA pathway, facilitated NB cell motility and invasiveness [49]. Poor survival, metastasis, and relapse have been associated with high serum concentrations of neuropeptide Y in NB [50]. The release of neuropeptide Y and neuropeptide Y/neuropeptide Y receptor 5 expression were favored in NB cells after treatment with the brain-derived neurotrophic factor (BDNF); the expression of the BDNF receptor, named tropomyosin-related kinase B receptor, has been associated with a worse prognosis in NB [43,44]. Neuropeptide Y receptor 5 antagonists inhibited the prosurvival actions induced by BDNF via the p44/p42 mitogen-activated protein kinase and phosphatidylinositol 3-kinase/Akt pathways, favored apoptosis, and sensitized resistant NB cells to chemotherapy [43,44]. The neuropeptide Y receptor 2 antagonist, BIIE0246, blocked the activation of the p44/p42 mitogen-activated protein kinase pathway, reduced cell proliferation, and, mediated by Bim, favored apoptosis. These effects were also found with neuropeptide Y/neuropeptide Y receptor 2 small interfering RNA [40]. BIIE0246 exerted anti-angiogenic effects by reducing the proliferation of endothelial cells expressing the neuropeptide Y receptor 2; antagonists of this receptor, but not neuropeptide Y receptor 5 antagonists, reducing vascularization [40,42,43].

Pro-neuropeptide Y processing has been associated with inferior outcomes/clinically advanced stages in NB [51]. Neuropeptide Y gene expression was decreased in SH-SY5Y NB cells after treatment with retinoic acid, favoring the pro-neuropeptide Y procession to neuropeptide Y [52]. The neuropeptide Y receptor 2 mediated glycolysis in NB cells, and this process was essential for these cells to obtain ATP under hypoxia circumstances [53]. Finally, neuropeptide Y inhibited calcium channel currents in SH-SY5Y NB cells [54].

#### 2.1.6. Neurotensin/Neuromedin U

Both peptides have been associated with the overall survival rate of patients with NB; neurotensin and neuromedin U levels were high in stage 4 NB individuals in which a less favorable outcome, in comparison with that observed in stage 4S NB patients, has been reported [55]. Both peptides favored the proliferation and invasion of SK-N-BE (2) NB cells, and it has been suggested that neurotensin and neuromedin U play an important role in regulating the tumor microenvironment, favoring NB development [55].

#### 2.1.7. Oxytocin

Oxytocin increased the number of SK-N-SH and SY-SY5Y NB cells, as well as the viability of the latter cells, without the activation of neurotrophic factors (neurotrophic growth factor, brain-derived neurotrophic factor); in addition, cell death promoted by hydrogen peroxidase was not counteracted with oxytocin; hence, it did not exert protective effects against oxidative stress mechanisms [56]. Glutamate had a toxic action on cell viability/proliferation and blocked neurite development in SH-SY5Y NB cells, but oxytocin counteracted the latter effect, exerted anti-apoptotic and protective actions, and augmented cell viability [57]. Oxytocin promoted neurite outgrowth by inhibiting calcium voltage-gated channels in SH-SY5Y NB cells and increased the gene expression of SHANK 1 and 3 proteins and the intracellular calcium concentration in these cells [58]. Oxytocin promoted changes in the cytoskeleton of SK-N-SH NB cells, and the peptide did not affect nerve growth factor/brain-derived neurotrophic factor gene expressions, but increased mRNA levels for both factors were observed when these cells were treated with oxytocin and atosiban, an oxytocin receptor antagonist [59]. This antagonist decreased oxytocin receptor mRNA levels and increased MAP 2 gene expression [59]. A dense F-actin filament recruitment was reported in the filopodia apical parts of SK-N-SH NB cells treated with oxytocin [59]. Oxytocin favored non-coding RNA tumor marker expression (metastasis-associated lung adenocarcinoma transcript 1 (MALAT 1)) in SK-N-SH NB cells and the cyclic adenosine monophosphate-responsive element-binding protein bound to the proximal promoter of the *MALAT 1* gene [60].

#### 2.1.8. Prolactin-Releasing Peptide

This peptide binds to the GRP10 receptor [61]. PrRP31, a prolactin-releasing peptide analog, upregulated the extracellular signal-regulated kinase/cyclic adenyl monophosphate response element-binding protein and phosphoinositide-3 kinase-protein kinase B/Akt signaling pathways, promoting SH-SY5Y cell growth and survival [61].

#### 2.1.9. sPEP1

sPEP1, a small 51-amino acid peptide, favored the self-renewal and aggressiveness of NB stem cells and exerted oncogenic actions by promoting the interaction of the SMAD family member 4 (SMAD4) and the eukaryotic translation elongation factor 1 alpha 1 (eEF1A1) [62]. sPEP1 interacted with the latter factor facilitating its binding to SMAD4, leading to SMAD4 transactivation repression and transcriptional upregulation of stem cell genes involved in tumor progression [62]. sPEP1 knockdown suppressed the in vivo metastasis and self-renewal of NB stem cells, and high levels of eEF1A1 or sPEP1 in the tissues of patients suffering from the disease have been related to poor survival [62]. Thus, sPEP1, via eEF1A1-suppressed SMAD4 transactivation, favored aggressiveness and self-renewal of NB stem cells.

#### 2.1.10. Substance P

Substance P, via the neurokinin-1 receptor, promoted the proliferation of SKN-BE (2) NB cells, whereas cell growth was blocked when these cells were treated with L-733,060, a neurokinin-1 receptor antagonist [63,64]. The presence of neurokinin-1 receptors has also been reported in other NB cells (Kelly, SH-SY5Y, SK-N-AS, SK-N-BE, IMR-5), and, after targeting this receptor, cell viability was decreased, and apoptotic mechanisms were induced [65]. Neurokinin-1 receptor inhibition promoted TP53 signaling and suppressed the transcription factor E2F2 in NB cells, and aprepitant (a neurokinin-1 receptor antagonist) reduced the tumor size in vivo [65]. Septide (a neurokinin-1 receptor agonist) decreased 6-hydroxydopamine-induced toxicity and blocked apoptosis by triggering pro-survival signal pathways (protein kinase B/Akt) in SH-SY5Y NB cells [66]. NB cells (CHP-212, SH-SY5Y) expressed neurokinin-1 and -2 receptors and preprotachykinin I. Both receptors mediated the proliferation of NB cells; truncated and full-length neurokinin-1 receptors were observed in these cells; and neurokinin-1 receptor suppression decreased the proliferation of NB cells [67]. Moreover, the expression of neurokinin-1 receptors, studied in 59 patients, was independent of the NB stage and biology [68].

P2X7, a receptor for extracellular nucleotides, was expressed in NB cells (e.g., SK-SY5Y, SK-N-BE (2), LAN-1, Lan-5) and in NB tissues [69]. The activation of this receptor caused NB cell proliferation but not apoptosis/caspase 3 activation; the silencing of the P2X7 receptor led to pro-apoptotic mechanisms; and NB cell growth was mediated by the secretion of substance P from NB-activated cells by nucleotides [69].

#### 2.1.11. Thyrotropin-Releasing Hormone

Thyrotropin-releasing hormone and three analogs (CG-3703 (montirelin), Z-thyrotropin-releasing hormone, and RGH-2202) exerted anti-apoptotic effects in SH-SY5Y NB cells, which were mediated by Bcl-2 protein (an increase was reported) and the activation of the phosphatidylinositol 3-kinase/Akt survival signaling pathway [70]. In fact, phosphatidylinositol 3-kinase inhibitors (LY-294002, wortmannin) counteracted the protective action exerted by thyrotropin-releasing hormone/RGH-2202 on tumor cells, but this action was not observed when PD-98059/U0126 (extracellular signal-regulated kinase 1/2/mitogen-activated protein kinase 1/2 inhibitors) were administered [70].

#### 2.1.12. Vasopressin

Tolvaptan, a vasopressin receptor 2 antagonist, decreased the proliferation and invasion of SK-N-AS NB cells expressing such receptor, promoting apoptosis [71]. The antagonist decreased pAkt/Akt ratio, the catalytic alpha subunit of protein kinase A and cyclic adenosine monophosphate levels; diminished collagenase type IV activity and anchorage-independent growth; and counteracted the ROCK1/2-RhoA pathway involved in controlling cell movement [71]. Raloxifene, an estrogen receptor modulator, reduced vasopressin mRNA levels and increased phosphorylated extracellular signal-regulated kinase 1/2 levels in SH-SY5Y NB cells [72]. MAPK/extracellular signal-regulated kinase 1/2 or protein kinase C inhibitors blocked the actions mediated by raloxifene [72]. The data show that vasopressin expression is regulated by raloxifene through G protein-coupled estrogen receptor 1 by activating protein kinase C and the mitogen-activated protein kinase/extracellular signal-regulated kinase pathway. Moreover, estradiol augmented vasopressin mRNA levels in SH-SY5Y NB cells, and this was also observed with estrogen receptor alpha agonists, but it was counteracted with estrogen receptor alpha antagonists (ICI 182,780) [73]. Estrogen receptor beta agonists reduced vasopressin expression, and estrogen receptor beta antagonists increased the action of estradiol on the expression of vasopressin [73]. Testosterone increased vasopressin expression in SH-SY5Y NB cells. This effect was blocked with estrogen receptor alpha antagonists, whereas estrogen receptor alpha agonists decreased the expression of the peptide [73]. Thus, testosterone and estradiol controlled vasopressin expression via estrogen receptors, exerting an inhibitory effect, through the estrogen receptor beta or a stimulatory action via the estrogen receptor alpha [73]. Finally, the stimulation of Akt mediated apoptosis in NB cells expressing a C98X vasopressin mutant following autophagy suppression [74]. The study in question also reported that autophagy-mediated degradation of mutated C98X peptides is needed because in the event that these peptides were not degraded, they will become toxic to cells [74]. Thus, Akt stimulation counteracted the autophagy beneficial actions, favoring cell death.

### 2.2. Anticancer Peptides

#### 2.2.1. Adrenomedullin/Pro-Adrenomedullin N-Terminal 20 Peptide

Both adrenomedullin/pro-adrenomedullin N-terminal 20 peptide (PAMP) blocked DNA synthesis and cell growth in human TGW NB cells [75]. Calcitonin gene-related peptide antagonists blocked the effects mediated by adrenomedullin but not those induced by PAMP. The effects mediated by adrenomedullin were also counteracted using adrenomedullin antagonists [75]. PAMP inhibited NB cell growth by blocking calcium channels (N-type) via pertussis toxin-sensitive G protein-coupled receptors since pertussis toxin blocked the effects mediated by PAMP [75]. This mechanism was different from that involved in the cell growth inhibition promoted by adrenomedullin since pertussis toxin did not block the effects mediated by the peptide [75].

A high concentration of adrenomedullin has been reported by radioimmunoassay in human NB tissues [76], and this peptide promoted the release of nitric oxide from human SK-N-SH NB cells by regulating the mobilization of intracellular free calcium [77]. The latter was blocked with adrenomedullin receptor antagonists or nitric oxide synthase inhibitors. Adrenomedullin favored calcium influx via L-type calcium channels and ryanodine-sensitive calcium release from the endoplasmic reticulum, and it seems that this occurred through cAMP-protein kinase A-dependent processes [77]. The increased concentration of calcium led to nitric oxide synthase activation and, finally, the release of nitric oxide. NB69 and IMR-32 NB cells increased the expression of adrenomedullin during hypoxia, whereas the expression of the receptor activity modifying protein 2 was reduced in IMR-32 cells but was not changed in NB69 cells under hypoxia [78]. Adrenomedullin and calcitonin gene-related peptide (CGRP) interacted with the same CGRP receptor expressed in SK-N-MC NB cells; in adrenomedullin, but not in CGRP, the N-terminal ring organization is crucial for its binding to the CGRP receptor [79].

#### 2.2.2. Corticotropin-Releasing Factor

Corticotropin-releasing factor/urocortin, after binding to the corticotropin releasing factor receptor 1, decreased both motility and proliferation in human SK-N-SH NB cells and favored the neuronal-like differentiation of these cells [80]. It seems that these mechanisms occurred after the activation of the cyclic adenosine monophosphate (cAMP)/protein kinase A/cAMP- responsive element binding protein pathway leading to reduced c-myc mRNA accumulation and increased levels of retinoblastoma protein and p27 (Kip1). Moreover, it has been reported that phospholipase C is not involved in the signaling pathways activated after the stimulation of corticotropin-releasing factor receptors in human SK-N-MC NB cells [81].

#### 2.2.3. Exendin-4

The peptide exendin-4, a glucagon-like peptide-1 receptor agonist, promoted a more differentiated phenotype, counteracted anchorage-independent growth, increased cell adhesion, and decreased cell migration in NB cells (SK-N-AS, SH-SY5Y) [82].

#### 2.2.4. Gonadotropin-Releasing Hormone

The activation of the gonadotropin-releasing hormone receptor blocked the growth of rat B35 NB cells, and apoptotic markers were expressed in these cells when they were treated with gonadotropin-releasing hormone [83]. Tumor xenograft growth (B35 NB cells) was slowed after gonadotropin-releasing hormone treatment [83].

#### 2.2.5. Orexin

Orexin A/orexin mRNA have been reported in human NB tissues [84]. Orexins (A and B), through the orexin receptor 1, inhibited SK-N-MC NB cell growth by favoring apoptotic mechanisms [85], and the blockade of TRPC3/6 channels disrupted the orexin receptor 1 signaling via NCX in IMR32 NB cells [86]. In SH-SY5Y NB cells, orexin A showed a protective effect against 6-hydroxydopamine-induced toxicity by exerting anti-apoptotic and anti-oxidant effects and by decreasing cell death markers. This protective effect was mediated through phosphatidylinositol 3-kinase and protein kinase C signaling pathways [87,88]. SB-3344868 (orexin receptor 1 antagonist), LY-294002 (phosphatidylinositol 3-kinase inhibitor), and chelerythrine (protein kinase C inhibitor) blocked the protective actions mediated by orexin A in SH-SY5Y NB cells against 6-hydroxydopamine [88]. Orexin A also protected SH-SY5Y NB cells against H_2_O_2_-induced damage (viability/superoxide dismutase decrease, apoptosis) through the phosphatidylinositol 3-kinase/extracellular signal-regulated kinase 1 and 2/mitogen-activated protein kinase 1/2 signaling pathways [89]. Moreover, orexin A, by blocking endoplasmic reticulum stress-induced apoptosis through the combination of phosphatidylinositol 3-kinase and Gi signaling pathways, protected SH-SY5Y NB cells against oxygen–glucose deprivation/reoxygenation-induced processes [90].

#### 2.2.6. SHPRH-146aa

circ-SHPRH (a unique circular RNA) and its derived peptide (SHPRH-146aa) are involved in NB pathogenesis and are promising therapeutic targets for NB treatment [91,92]. circ-SHPRH was overexpressed in NB samples, and this overexpression, like that of the SHPRH-146aa peptide, blocked the proliferation, migration, and invasion of NB cells (SK-N-AS, SH-SY5Y, SK-H-SH, IMR-32). Increased apoptosis was also reported, as well as caspase upregulation and Bcl-2 downregulation [92]. SHPRH-146aa peptide counteracted the malignancy traits of NB cells, and a SHPRH-146aa–RUNX1 protein interaction was found, leading to an increased expression of the protein coding gene NFKBIA. This interaction suggests a new pathway controlling apoptotic mechanisms in NB cells [92]. NFKBIA knockdown blocked the pro-apoptotic actions mediated by SHPRH-146aa on NB cells. circ-SHPRH expression in NB tissues and cell lines (SH-SY5Y, SK-N-BE (2)) has been studied [91]. circ-SHPRH upregulated the crucial cell cycle regulator p21 expression to block cyclin-dependent kinases in NB, and circ-SHPRH overexpression limited NB tumor growth in NOD scid gamma mice [91]. Figure 1 shows the peptidergic systems involved in NB development.

### 2.3. Other Peptides and Neuroblastoma

#### 2.3.1. Adrenocorticotropin Hormone

Olfactory NB (esthesioneuroblastoma), a rare neuroectodermal tumor, arises from the olfactory neuroepithelium, and the release of adrenocorticotropin hormone (ACTH) from this tumor has rarely been observed; however, olfactory NB can cause ectopic ACTH syndrome (even long after the initial clinical treatment) because the presence/release of ACTH from olfactory NB cells has been reported [93,94,95,96]. Serum ACTH/cortisol levels were reduced after olfactory NB surgical removal [95], and it has been suggested that olfactory NB stimulation (e.g., chemotherapy, biopsy) triggers the onset of ectopic ACTH syndrome [94].

#### 2.3.2. Melanin-Concentrating Hormone

Human NB tissues and cell lines (IMR-32, NB69) expressed melanin-concentrating hormone (MCH) receptor mRNA [97,98], whereas human Kelly NB cells expressed MCH and MCH receptor 1 but not MCH receptor 2 [99]. In the latter cells, MCH did not increase the level of free intracellular calcium but induced mitogen-activate protein kinase (MAPK) phosphorylation. This suggests that MCH acts via Galpha(i)/Galpha(o) in Kelly NB cells [99]. MCH, via MAPK and p53 signaling pathways, promoted neurite outgrowth in human SH-SY5Y NB cells [100]. These cells expressed MCH receptor 1 mRNA/protein, and the peptide regulated potassium currents, favored MAPK phosphorylation, and promoted the expression and nuclear localization of phosphorylated p53 protein. The peptide also augmented Elk-1 phosphorylation, upregulated Egr-1 expression (Elk-1 and Egr-1 are transcriptional factors), and favored neurite outgrowth [100]. The results suggest that MCH is involved in neuronal differentiation.

#### 2.3.3. Neuropeptide FF

SH-SY5Y NB cells expressed neuropeptide FF receptor 2 [101]. The extracellular signal-regulated protein kinase pathway was activated after treatment with neuropeptide FF in these cells. Moreover, protein kinase A/nitric oxide synthases were involved in this activation, and neuropeptide FF also activated the NF-κB pathway [101]. SK-N-MC NB cells expressed both neuropeptide FF receptor 2 and neuropeptide YY [102]. The activation of this receptor by agonists led to the activation of the MAPK signaling pathway, actin cytoskeleton reorganization occurred, and the expression of neuropeptide FF receptor 2 mRNA/protein was upregulated by neuropeptide FF [102]. Other studies have reported that neuropeptide FF analogs antagonized opioid activities in neuropeptide FF receptor 2-transfected SH-SY5Y NB cells expressing mu/delta opioid receptors [103], and this antagonism was also observed in the same NB cell line regarding the neuropeptide FF receptor 1 [104]. The activation of the latter receptor blocked voltage-gated calcium current (N-type) and the activity of adenylyl cyclase but increased the release of intracellular calcium mediated by the stimulation of muscarinic receptors [104]. Thus, neuropeptide FF, via neuropeptide FF receptors 1/2, exerted anti-opioid effects; in addition, an interaction between neuropeptide FF and opioid receptors occurred. Moreover, peptides (SQA-neuropeptide FF, neuropeptide AF, neuropeptide FF) derived from the neuropeptide FF precursor were located in SH-SY5Y NB cells [105].

#### 2.3.4. Somatostatin

A review focused on the current status of, and future perspectives on, nuclear medicine imaging and the treatment of NB has been published [106]. Olfactory NB cells express somatostatin receptor 2, which is a target for radionuclide imaging, for example, with ^68^Ga-DOTATATE [107,108,109]. A study has reported that 75% of patients with olfactory NB expressed somatostatin receptor 2, and 7.5% of them expressed somatostatin receptor 5 [108]. Moreover, the expression of the somatostatin receptor 2 has been also reported in NB tissue [110]. The therapeutic potential of combined chemotherapy and peptide receptor radionuclide therapy to treat refractory/relapsed metastatic NB has been suggested; however, the results are preliminary; hence, a more in-depth investigation is required [111]. Children with relapsed NB do not respond to I-meta-iodobenzylguanidine (MIBG) therapy, and other radiolabeled agents (e.g., DOTA-conjugated peptide) have been suggested to treat these patients expressing somatostatin receptors 2 [112].

Radiopharmaceuticals (^177^Lu-octreotide, ^177^Lu-octreotate) have been used for the treatment of human NB cell (CLB-BAR)-bearing mice [113]. After treatment with these somatostatin analogs, pro-apoptotic (TRADD, CASPS, TNSF8/10) and anti-apoptotic (IL10) genes were regulated. The antitumor effects observed were better with ^177^Lu-octreotide treatment than with ^177^Lu-octreotate [113]. The data confirm the activation of apoptotic mechanisms mediated by both somatostatin analogs and highlight the beneficial use of radiopharmaceuticals in nuclear medicine to detect and treat NB. The therapeutic potential of ^177^Lu-octreotate in NB xenograft models (BALB/c nude mice bearing IMR-32, CLB-GE, CLB-BAR tumor xenografts) has also been demonstrated, and the use of this radiopharmaceutical to fight metastatic NB has been suggested [114].

#### 2.3.5. Vasoactive Intestinal Peptide/Pituitary Adenylate Cyclase-Activating Polypeptide

VIP level increased during NB differentiation [115], and NB can regress to a benign form or advance to a malignant metastatic tumor [116]. Vasoactive intestinal peptide (VIP) and PACAP promoted NB cell differentiation into a benign form by controlling vascular endothelial growth factor, vascular endothelial growth factor receptor, and hypoxia-inducible factor expressions [116]. The study showed that VIP and PACAP interfered with the cell differentiation regulating angiogenic and hypoxic mechanisms. This is important since hypoxia favored the activation of hypoxia-inducible factors, which promoted cell proliferation and metastasis and the release of the vascular endothelial growth factor [116]. VIP augmented dopamine transporter and synaptic vesicle glycoprotein 2C in IMR-32 NB cells [117].

Pituitary adenylate cyclase-activating polypeptide (PACAP), after binding to PAC-1 receptors, promoted its own expression in human NB-1 NB cells. This process was mediated by protein kinase A, extracellular signal-regulated kinase, and novel protein kinase isoforms [118]. NB-1 NB cells also expressed VPAC2 receptors [119], and PACAP, via the PAC1 receptor, favored VIP/fos gene expressions in these cells [120]. PACAP-38, via PAC-1 receptors, favored neuronal differentiation in SH-SY5Y NB cells through the cyclic adenyl monophosphate-induced activation of p38 mitogen-activated protein kinase and extracellular signal-regulated kinase pathways [121]. PACAP-38 did not promote the proliferation of SH-SY5Y NB cells; however, the leukemia inhibitory factor favored such proliferation, and SH-SY5Y cells treated with PACAP-38 increased Bcl-2 expression as well as resistance to hypoxic situations [122]. It seems that PACAP-27 protected murine neuro2a NB cells from apoptotic mechanisms, whereas VIP had no effect [123]. PAC-1 receptor-mediated neurite outgrowth and cytoprotection processes in murine neuro2a NB cells. These mechanisms were transglutaminase (TG2)-dependent [124]. Cytoprotection and TG2 activation were also observed in human SH-SY5Y NB cells, PACAP-27 augmented the activity of TG2, and the blockade of TG2 inhibited the PACAP-27-promoted attenuation of neurite outgrowth and hypoxia-induced cell death [124].

The effects of VIP/PACAP27/PACAP-38 and VIP/PACAP analogs on Mycn-amplified SK-N-DZ/IMR-32 NB cells and Kelly NB cells—presenting, in addition, a mutation (F1174L ALK)—have been studied [115]. The three NB cell lines expressed genes encoding the receptors (PAC-1, VPAC-1, VPAC-2) for VIP and PACAP. VIP promoted neuritogenesis in Kelly/SK-N-DZ cells and reduced Akt activity and Mycn expression in Kelly cells. These actions were dependent of protein kinase A [115]. Moreover, VIP blocked IMR-32/Kelly cell invasion [115]. The peptidergic systems involved in NB are shown in Table 1.

## 3. Perspectives and Future Research

Peptides such as amylin, angiotensin II, bradykinin, gastrin-releasing peptide, bombesin, neuropeptide Y, neurotensin, neuromedin U, prolactin-releasing peptide, oxytocin, sPEP1, thyrotropin-releasing hormone, and substance P, after binding to their respective receptors (e.g., GRP receptor, neuropeptide Y receptor 2/5, GRP10 receptor), favored NB progression (tumor cell proliferation, migration and invasion, anti-apoptotic effect, angiogenesis) [15,17,21,25,28,29,36,37,40,42,43,55,56,57,61,62,63,70]. This means that the use of peptide receptor antagonists is a promising anti-NB research line that must be fully developed because NB cells express/overexpress peptide receptors (e.g., neuropeptide Y, GRP), and antagonists of these receptors counteract NB growth (e.g., GRP receptor antagonists). On the contrary, other peptides exert anticancer effects (cell proliferation/migration inhibition, apoptosis) against NB. This is the case of adrenomedullin, PAMP, corticotropin-releasing factor, urocortin, exendin-4, gonadotropin-releasing hormone, orexin, and SHPRH-146aa peptide [75,80,82,83,85,92]. Accordingly, the therapeutic potential of combined peptide receptor radionuclide therapy and chemotherapy to fight NB has been proposed [111]. However, despite the existing data, basic research is needed in NB since, as mentioned in the previous section, the proliferative/antiproliferative and migration/antimigration activities exerted by many peptides (e.g., adrenocorticotropin hormone, MCH, neuropeptide FF, somatostatin, opioids, VIP, and PACAP) is unknown, as is whether NB cells synthesize and release peptides exerting autocrine proliferative actions and anti-apoptotic mechanisms, favoring NB cell viability and survival. The expression of many types of peptide receptors is currently unknown. It is also unknown as to whether peptide receptor antagonists elicit an antitumor action or not, as well as the tissue/serum concentrations of many peptides. This is important because a high level of peptides (e.g., neurotensin, neuromedin U) has been associated with a less-NB-favorable outcome [55], and peptide receptor overexpression (e.g., GRP receptor) favored NB cell migration and invasion [28]. Although some peptides (GRP, neuropeptide Y) have been more-thoroughly studied, unfortunately, much of the existing data regarding the above-mentioned aspects are fragmentary, scarce, or absent in many peptidergic systems; hence, there is still a lot of basic research work to be conducted in NB. The expression of peptide receptors is important since compared with that reported in benign tumors, an increased receptor expression (e.g., GRP receptor) was found in more aggressive and undifferentiated NB [25]. Poor survival, metastasis, and relapse have been related with a high serum peptide level (e.g., neuropeptide Y) in NB [50]. A high tissue level of sPEP1 has been associated to NB poor survival [62], and peptide precursor processing (e.g., pro-neuropeptide Y) has also been related in NB with clinically advanced stages and inferior outcomes [51]. To develop future anti-NB strategies, it must be taken into consideration that the expression of the same peptidergic receptor (e.g., ATR1) has been reported in some NB cell lines, but in others, this expression was not observed [16], and it was found that human NB cells have different biological features expressing different peptides [39].

A crucial research line is to determine the how peptidergic systems (peptides and receptors) are regulated. Currently, it is known that VIP augments ATR1 density in NB cells [16]; that insulin/insulin-like growth factor-1 potentiate the proliferative action mediated by angiotensin II on NB cells [17]; that BDNF, via the tropomyosin-related kinase B receptor, promotes the release of proliferative peptides/peptide receptor expression (e.g., neuropeptide Y and neuropeptide Y receptor 5, a high expression of this receptor has been related with a worse NB prognosis) [43,44]; that the retinoic acid decreases neuropeptide Y gene expression in NB [52]; and that the synthesis/release of peptides (e.g., neuropeptide Y) from NB cells are mediated muscarinic receptors [45,46]. The estrogen receptor modulator raloxifene decreases vasopressin mRNA levels in NB cells [72], and estradiol/testosterone increase such levels in the same cells [73]. CGRP antagonists block the anticancer effects mediated by adrenomedullin but not those induced by PAMP [75], and PACAP, via the PAC-1 receptor, favors its own expression in NB cells [118]. This is an important finding that merits further investigation in other peptidergic systems because PACAP favors NB cell differentiation into a benign form [116]. There is still much to do and much to learn; this research line is important and must be developed since by knowing the factors that regulate the synthesis and release of oncogenic or anticancer peptides, NB development could be controlled. It is known that several peptides originate from the same precursor; therefore, another important line of research is to understand how the gene expression of this precursor and the different peptides that originate from it are regulated in NB cells.

As mentioned in the previous section, a plethora of anti-NB strategies involving the peptidergic systems can be applied alone or in combination therapies. The beneficial actions exerted by the oncogenic peptides on NB development can be inhibited by using the following: (1) Peptide receptor antagonists: PD123319 (ATR2 antagonist) blocks the effects mediated by angiotensin II [20]. RC-3095 (GRP receptor antagonist) counteracts tumor growth/angiogenesis mediated by bombesin, increases pro-autophagic protein expression in NB cells, and decreases GRP release from NB cells and NB cell growth [32,33]. It is important to note that a low concentration of RC-3095 blocks murine NB cell proliferation (this was inhibited with a histone deacetylase inhibitor, suggesting the involvement of epigenetic mechanisms), but a high concentration promotes murine NB cell proliferation [33]. H2756 (GRP receptor antagonist) blocks Akt phosphorylation mediated by bombesin in NB cells [27]. Neuropeptide Y receptor 5 antagonists block the prosurvival actions mediated by BDNF, induce apoptosis, and sensitize resistant NB cells to chemotherapy [43,44]. BIIE0246 (neuropeptide Y receptor 2 antagonist) decreases NB cell proliferation, favors apoptosis, and promotes anti-angiogenic actions) [40,42,43]. The proliferative action promoted by insulin on NB cells is inhibited with the angiotensin II receptor antagonists DuP753 (ATR1) or PD123177 (ATR2) [17]. The oxytocin receptor antagonist atosiban reduces oxytocin receptor mRNA levels [59]. L-733,060 (neurokinin-1 receptor antagonist) induces apoptosis in NB cells [63,64]. Aprepitant (neurokinin-1 receptor antagonist) decreases NB tumor size [65]. And the vasopressin receptor 2 antagonist tolvaptan reduces the proliferation/invasion of NB cells and also promotes apoptosis [71]. (2) Purinergic P2X7 receptor antagonists block the proliferative action regulated by bradykinin [21]. (3) Anti-GRP antibodies inhibit the NB growth mediated by GRP [1,25]. (4) Integrin β1 silencing blocks NB cell migration; GRP receptor overexpression upregulates integrin α2/α3/β1 protein/mRNA expressions and favors NB cell migration [28]. (5) Phosphatidylinositol 3-kinase signaling pathway inhibitors (e.g., LY-294002, wortmannin) counteracts the Akt phosphorylation mediated by bombesin in NB cells [27] and the protective effects promoted by thyrotropin-releasing hormone on NB cells [70]. (6) Y15 (FAK inhibitor) blocks NB cell growth/metastasis mediated by GRP; NB cell growth is favored by FAK overexpression [30]. (7) GRP receptor knockdown in NB cells: cell proliferation/invasion/metastasis decrease, mRNA level decrease in oncogenes regulating NB progression, angiogenesis inhibition, PTEN expression upregulation (inhibitor of the phosphatidylinositol 3-kinase/Akt signaling pathway; a tumor suppressor), Akt downregulation, liver metastasis reduction, and aerobic glycolysis blockade [24,26,31,34]. (8) sPEP1 knockdown blocks self-renewal and metastasis of NB stem cells [62]. (9) The acetic acid analog dichloroacetate decreases NB cell proliferation and favors NB cell death, and dichloroacetate and GRP receptor silencing synergistically inhibit the proliferation of these cells [31]. (10) The transcription factor Ets 1 regulates GRP receptor transcription in NB cells; thus, by downregulating its expression, the mitogenic effect mediated by GRP on NB cells could be blocked [35]. (11) Anticancer peptides (adrenomedullin, PAMP, corticotropin-releasing factor, urocortins, exendin-4, gonadotropin-releasing hormone, orexin, SHPRH-146aa peptide) could be administered to counteract the effects exerted by the oncogenic peptides [75,80,82,83,85,92]. Many of the possible therapeutic strategies against NB must be confirmed and developed in the future; in many cases, the current data are preliminary, fragmentary, scarce, or absent. Figure 2 summarizes the anti-NB therapeutic strategies.

## 4. Conclusions

Surgical resection, chemoradiation, phosphoinositide 3-kinase inhibitors, autologous stem cell rescue, immunotherapy, cyclin-dependent kinase 3/6 inhibitors, and treatment with retinoic acid are some on the therapeutic strategies employed to fight NB [1,2,3,4,8]. In addition to the previous strategies, another has been focused on understanding how the peptidergic systems are involved in NB. In this sense, it is widely known that peptides are oncogenic and anticancer agents [11], and it is therefore a promising line of research that merits development for tumor diagnosis and treatment. In fact, some peptide drugs (e.g., gonadotropin-releasing hormone analogs, somatostatin analogs) have been approved by the FDA to fight some cancer types (e.g., breast, prostate, lung neuroendocrine), as well as for cancer diagnosis [125]. However, no antitumor drug involving the peptidergic systems is currently available in clinical practice to treat NB. This means that basic research (in vitro and in vivo experiments) is urgently needed to increase our knowledge concerning the involvement of peptides in NB and to transfer this knowledge into clinical practice. For example, although some peptidergic systems have been studied in NB, the oncogenic/anticancer effects (e.g., proliferative, antiproliferative, migration, apoptosis) mediated by other peptidergic systems have not yet been studied in NB and are currently unknown. It is important to understand, in depth, which peptides are synthetized and released from NB cells, the peptide receptors that express/overexpress these cells and whether the expression of these receptors is involved in the viability of NB cells, and the tissue/serum levels of peptides in NB. This is crucial because a high peptide concentration [55] and a peptide receptor overexpression [28] have, respectively, been related to a less-NB-favorable outcome and shown to favor NB cell migration and invasion. Moreover, this overexpression could be used as a tumor biomarker for NB. Peptide receptor overexpression in tumor cells is very useful for the diagnosis and treatment of tumors because this overexpression facilitates the delivery of chemotherapeutic drugs or compounds that promote apoptotic mechanisms in cancer cells; in fact, the therapeutic potential of combined chemotherapy and peptide receptor radionuclide therapy as an anti-NB strategy has been suggested [111]. Moreover, peptide receptor overexpression is important in developing anti-NB strategies using peptide receptor antagonists or agonists. Although some peptides (GRP, neuropeptide Y) have been more-thoroughly studied in NB, unfortunately, much of the data regarding this disease and the peptidergic systems are fragmentary, scarce, or absent; hence, there is still a lot of basic research work to be completed, and, in particular, in vivo experiments must be performed.

Oncogenic peptides promote NB progression, and this means that peptide receptor antagonists (PD123319, RC-3095, H2756, BIIE0246, DuP753, PD123177, L-733,060, aprepitant, tolvaptan) could be used as anti-NB agents since these compounds counteract NB growth and migration and angiogenesis. It is important to highlight that the same peptide (e.g., substance P) can bind to several G protein-coupled receptors, activating a variety of signaling pathways, which is crucial when studying the effect of a specific antagonist as an antitumor agent. This is an important point to investigate in NB. There are few studies on the use of peptide receptor antagonists in NB, and the number of antagonists tested is also scarce. Thus, studies on other peptide receptor antagonists, in addition to those already studied, to treat NB must be urgently developed in preclinical studies, and experiments using antagonists in combination with chemotherapy/radiotherapy must also be carried out that we might learn whether an anticancer synergic action occurs and whether the side-effects mediated by the cytostatics are reduced. Moreover, studies on P2X7 receptor antagonists, anti-oncogenic peptide antibodies, integrin β1 expression, signaling pathways involved and signaling pathway inhibitors (wortmannin, LY-294002), FAK inhibitors (Y15), peptide receptor knockdown, sPEP1 knockdown, acetic acid analogs (dichloroacetate), and the transcription factor Ets 1 must be fully developed to counteract the beneficial effects mediated by oncogenic peptides on NB. In the same way, the use of anticancer peptides to fight NB must be studied in depth; these peptides could counteract the effects mediated by the oncogenic peptides. However, peptides show a poor bioavailability, short half-life, and high solubility and safety. To increase the therapeutic option of anti-NB peptides, some strategies are currently available, such as amino acid sequence manipulation, peptide conjugation to polymers, peptide-loaded nanoparticles, peptide cyclization, cell-targeting peptides, and cell-penetrating peptides, to improve the delivery and stability of peptides [125,126]. The interactions between the peptidergic systems in NB must be investigated in depth. In this sense, it is crucial to understand the antitumor actions promoted by combining peptides or peptide receptor agonists exerting anti-NB actions and peptide receptor antagonists, as well as to determine the antitumor synergistic actions exerted by different peptide receptor antagonists favoring apoptotic mechanisms in NB cells. The involvement of oncogenic and anticancer peptides in NB progression shows the interactions between those peptides and the functional complexity of the mechanisms regulated by peptides in cancer development. Figure 3 shows the chemical structures of promising anti-NB effects.

The research line focusing on the mechanisms regulating the expression of oncogenic and anticancer peptides and their receptors in NB must also be potentiated. This is crucial because, for example, in NB cells, peptides control the expression of peptide receptors [16] and potentiate the effects mediated by other peptides [17]; some compounds favor the release of proliferative peptides and the expression of peptide receptors, as well as controlling the gene expression of peptides [43,44,52]; and peptides (PACAP) promote, after binding to their receptors, their own expression [118]. The latter must be highlighted because PACAP promotes the differentiation of NB cells into the benign form [116]. This research line must be potentiated in order to understand the mechanisms regulating both the synthesis and the release of oncogenic or anticancer peptides. This knowledge will assist in the development of therapeutic strategies to control NB progression. Moreover, the roles played by peptides in the tumor microenvironment must be elucidate, and it is especially important to understand how peptides control antitumor defenses by regulating the actions mediated by immune cells. Epigenetic mechanisms have been involved in the expression of peptide receptors, the recurrence rate, and carcinogenesis; thus, these mechanisms must also be studied in NB to understand how they control the peptidergic systems in NB.

In summary, many of the possible therapeutic strategies against NB must be confirmed and developed; in many cases, the current data are preliminary, fragmentary, scarce, or absent. However, peptidergic systems are potential and promising targets for the diagnosis and treatment of NB, and the current data suggest that drug cocktails including different compounds (anticancer peptides and peptide receptor antagonists) would be beneficial in treating the disease, either alone or in combination with conventional (radiotherapy/chemotherapy, surgery) or new (immunotherapy) anti-NB therapies. To develop clinical trials and increase the possibility of transversal research, basic research on the peptidergic systems in NB should be strengthened. Although the data regarding the involvement of the peptidergic systems in NB are, in many cases, fragmentary or very scarce for a particular peptidergic system, taken together, they are quite promising in terms of the potential development of this research line with the aim of developing new therapeutic strategies to treat NB in the future.

## Figures and Tables

**Figure 1 ijms-26-03464-f001:**
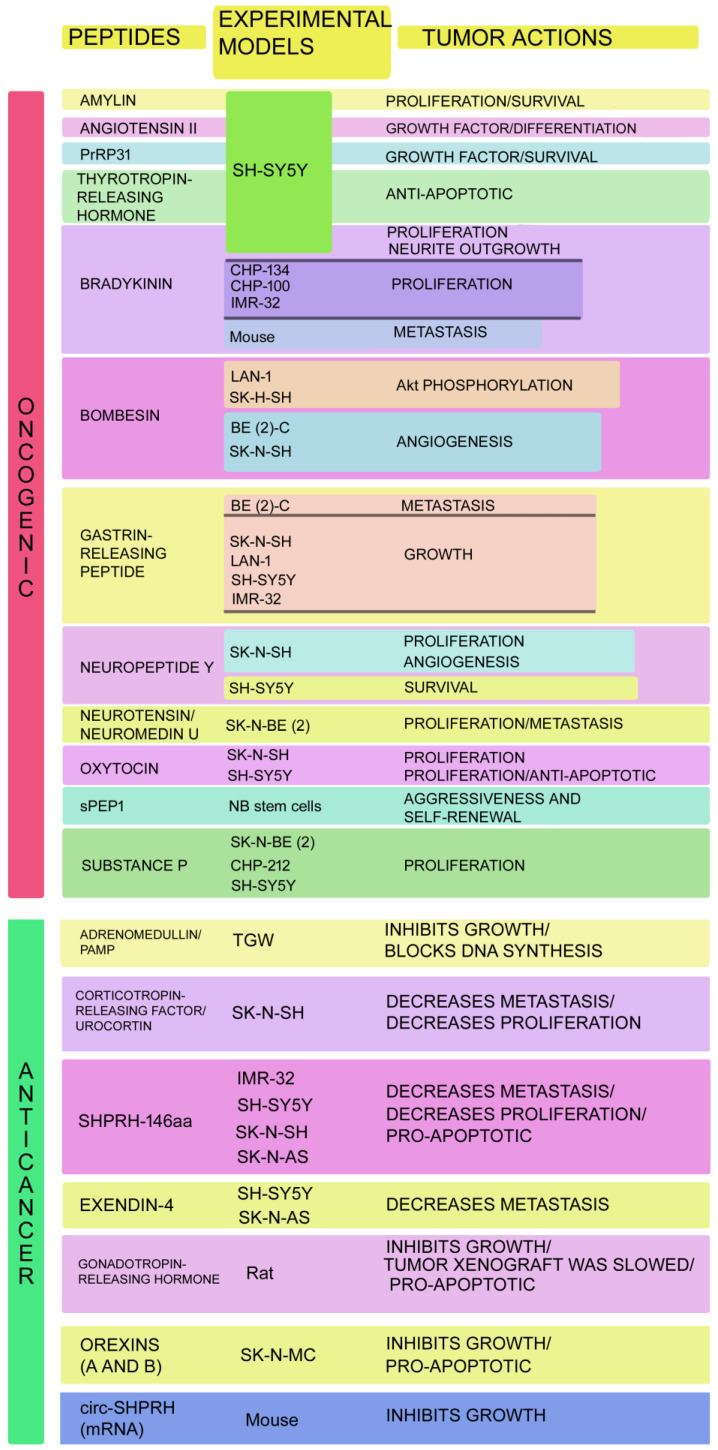
Oncogenic and anticancer peptides involved in NB. The experimental models used and the tumor effects mediated by these peptides are mentioned. Adrenomedullin/PAMP [75]; Amylin [15]; Angiotensin II [17,19]; Bombesin (Akt phosphorylation [27]; angiogenesis [33,37]); Bradykinin (proliferation and metastasis [21]; neurite outgrowth [22]); circ-SHPRH [91]; Corticotropin-releasing factor/urocortin [80]; Exendin-4 [82]; Gastrin-releasing peptide (metastasis [24,28,30]; growth [25]); Gonadotropin-releasing hormone [83]; Neuropeptide Y (proliferation [40,42,43]; angiogenesis [40,43]; survival [47]); Neurotensin/neuromedin U [55]; Orexins [85]; Oxytocin [56,57]; PrRP31 [61]; SHPRH-146aa [92]; sPEP1 [62]; Substance P [63,64,67]; Thyrotropin-releasing hormone [70].

**Figure 2 ijms-26-03464-f002:**
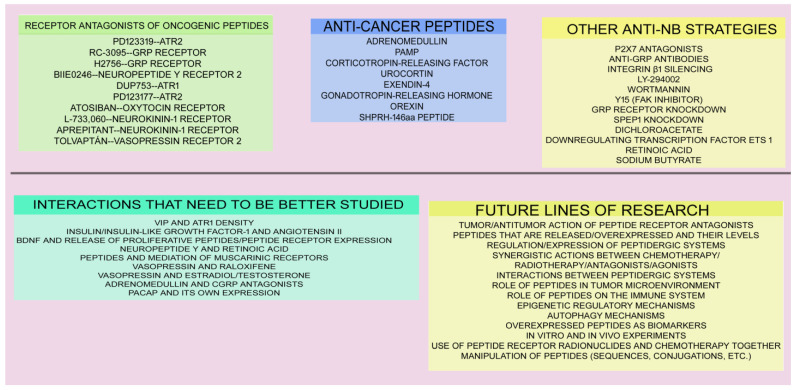
Anti-NB therapeutic strategies. Future research lines and interactions of the peptidergic system that must be investigated in depth are also mentioned.

**Figure 3 ijms-26-03464-f003:**
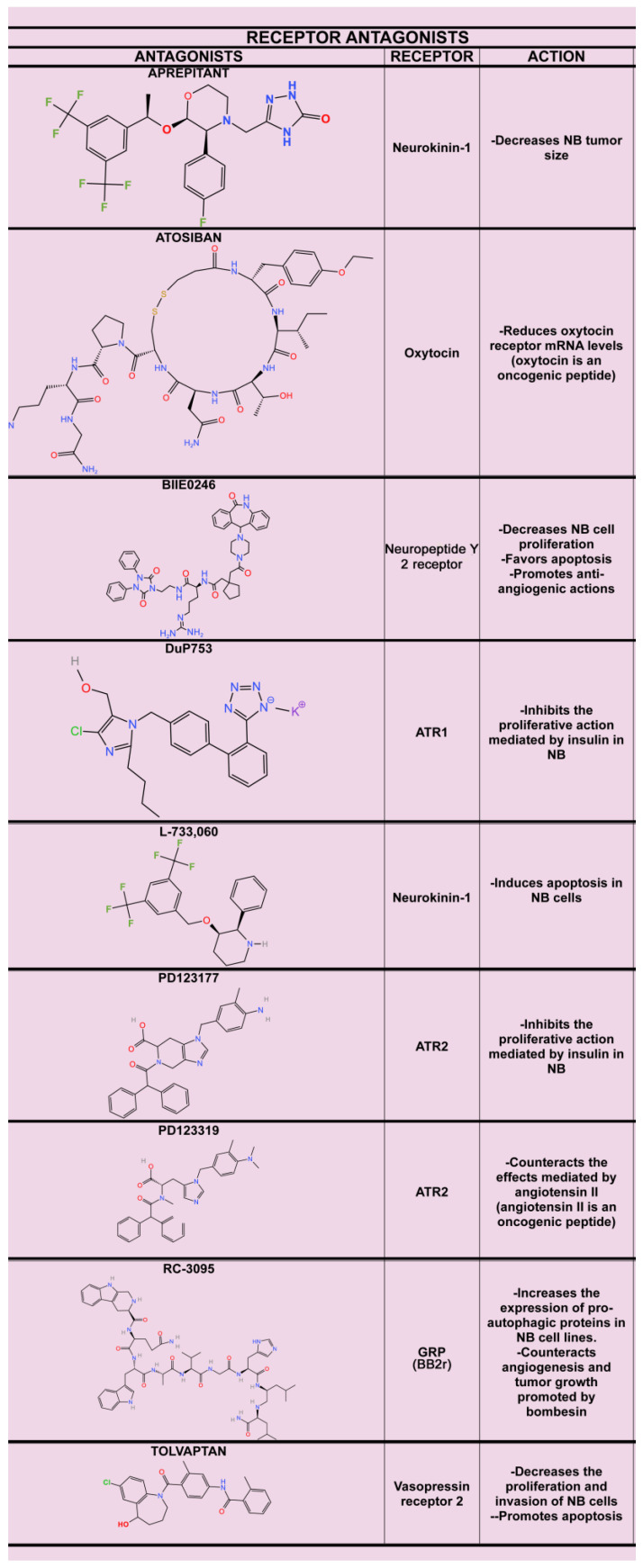
Chemical structures of peptide receptor antagonists exerting an anti-NB action. The receptors involved and the anti-NB actions exerted are also indicated. Aprepitant [1]; Atosiban [61]; BIIE0246 [42,44,45]; DuP753 [19]; L-733,060 [65,66]; PD123177 [40]; PD123319 [22]; RC-3095 [34,35,39]; Tolvaptan [71].

**Table 1 ijms-26-03464-t001:** Summary of the most important findings regarding the involvement of the peptidergic systems in NB.

Peptides	Effects on Neuroblastoma	References
Adrenocorticotropin Hormone (ACTH)	- Olfactory NB releases ACTH.	[93,94,95,96]
Adrenomedullin/Pro-Adrenomedullin N-terminal 20 Peptide (PAMP)	- Both peptides block DNA synthesis and cell growth in NB cells; - CGRP antagonists block the previous effects mediated by adrenomedullin but not those by PAMP.	[75]
Amylin	- Favors NB cell proliferation and survival; - Favors vascular endothelial growth factor/HspB5 release from endothelial cells.	[15]
Angiotensin II	- NB cells express ATR1, ATR2, and ATR 4 receptors; - VIP increases ATR1 density in NB cells; - Acts as a growth factor in NB cells: insulin and insulin-like growth factor-1 increase the proliferative effects mediated by angiotensin II; - Favors NB cell differentiation by increasing MAP2 level via AT2R.	[16,17,18,19]
Bradykinin	- Promotes NB cell proliferation and increases P2X7B receptor expression: bradykinin and purines are involved in NB dissemination, favoring metastasis; - Increases metalloproteinase activity, favors cell adhesion, and upregulates vascular endothelial growth factor expression in NB cells; - Favors neurite outgrowth mediated by the protein kinase C-dependent ROCK/RhoA pathway and protein phosphatase 2A.	[21,22]
Corticotropin-Releasing Factor (CRF)	- CRF/urocortin reduce motility/proliferation in NB cells and favor neuronal-like differentiation; - Previous mechanisms occur when the cAMP/protein kinase A/cAMP-responsive element binding protein pathway is activated.	[80]
Exendin-4	- Favors a more differentiated phenotype, counteracts anchorage-independent growth, increases cell adhesion, and decreases cell migration in NB cells.	[82]
Gastrin-Releasing Peptide (GRP)/Bombesin	- GRP acts as an autocrine growth factor in NB cells; - GRP is synthetized by NB cells; - GRP activates the phosphatidylinositol 3-kinase/Akt survival signaling pathway; - Phosphatidylinositol 3-kinase/Akt signaling pathway activation is correlated with poor prognosis; - GRP increases NB cell migration and matrix metalloproteinase-2 expression; - Integrin β1 silencing blocks NB cell migration mediated by GRP; - FAK overexpression favors NB cell growth and FAK blockers (Y15) inhibit NB cell growth and metastasis promoted by GRP; - GRP receptor overexpressed in NB: an augmented receptor expression was observed in more aggressive and undifferentiated tumors when compared with that found in benign tumors; - GRP receptor inhibition increases autophagy-mediated degradation of GRP in NB cells; - GRP receptor silencing blocks key regulators of aerobic glycolysis mechanisms; - GRP/GRP receptor/Akt2 axis is involved in NB progression; - Changes in cell morphology, reduced cell proliferation/size/migration, angiogenesis blockade, DNA syntesis inhibition, Akt downregulation, PTEN expression upregulation, and anchorage-independent growth suppression was observed in GRP receptor knockdown NB cells; - Anti-GRP antibodies inhibit NB cell proliferation and favors interferon γ/cytotoxic granzyme B release from natural killer cells; - PTEN/Akt signaling pathway is a key mediator of the GRP tumorigenic properties in NB cells; - Bombesin mediates NB cell growth and promotes angiogenesis; - Protein kinase C regulates bombesin-mediated secretion of the vascular endothelial growth factor in NB cells and when this secretion is blocked, a reduced NB cell proliferation-induced by GRP occurs.	[1,22,23,24,25,26,27,28,29,30,31,32,34,36,37]
Gonadotropin-Releasing Hormone (GRH)	- Activation of GRH receptors inhibits NB cell growth and apoptotic markers are expressed in these cells after treatment with GRH; - NB cell xenograft growth slowed after GRH treatment.	[83]
Melanin-Concentrating Hormone (MCH)	- NB tissues and cell lines express MCH and MCH receptor mRNA; - MCH, via MAPK and p53 signaling pathways, promotes neurite outgrowth in NB cells;	[97,98,99,100]
Neuropeptide FF	- NB cells express neuropeptide YY and neuropeptide FF receptor 2; - Peptides (SQA-neuropeptide FF, neuropeptide AF, neuropeptide FF), derived from the neuropeptide FF precursor, located in NB cells; - Activates the extracellular signal-regulated protein kinase and NF-κB pathways in NB cells; - Activates the mitogen-activated protein kinase signaling pathway, promotes actin cytoskeleton reorganization, and upregulates the expression of neuropeptide FF receptor 2 mRNA/protein; - Neuropeptide FF analogs antagonize opioid activities in NB cells expressing mu/delta opioid receptors.	[101,102,103,105]
Neuropeptide Y	- NB cells express neuropeptide Y and neuropeptide Y receptors 1, 2, 4, and 5; - Neuropeptide Y, released from NB cells, exerts an autocrine action favoring NB cell proliferation and angiogenesis; - Exerts anti-apoptotic mechanisms increasing the viability/survival of NB cells; - Favors NB cell motility and invasiveness; - Neuropeptide Y gene expression decreases in NB cells after treatment with retinoic acid; - Poor survival, metastasis, and relapse associated with high neuropeptide Y serum levels; - Neuropeptide Y release and neuropeptide Y/neuropeptide Y receptor 5 expression favored in NB cells after treatment with BDNF; the expression of the BDNF receptor, named tropomyosin-related kinase B receptor, has been associated with a worse prognosis; - Pro-neuropeptide Y processing associated with inferior outcomes/clinically advanced stages.	[39,40,41,43,44,48,49,50,51,52]
Neurotensin/Neuromedin U	- Both peptides favor NB cell proliferation and invasion; - High neurotensin and neuromedin U levels related to a less favorable outcome.	[55]
Orexin	- Orexin A/orexin mRNA in NB tissue; - Orexin A and B, via orexin receptor 1, block NB cell growth and favor apoptosis.	[84,85]
Oxytocin	- Favors proliferation/viability and anti-apoptotic mechanisms in NB cells; - Promotes neurite outgrowth and cytoskeleton changes; favors non-coding RNA tumor marker expression.	[56,57,58,59,60]
Prolactin-Releasing Peptide (PrRP)	- Binds to GRP10 receptors; - PrRP31 favors NB cell growth and survival.	[61]
sPEP1	- Favors self-renewal and aggressiveness of NB stem cells; - sPEP1 knockdown inhibits metastasis and self-renewal of NB stem cells;- sPEP1 high tissue levels related to poor survival.	[62]
SHPRH-146aa	- Its overexpression blocks NB cell proliferation, migration, and invasion; increases apoptosis; and conunteracts NB cell malignancy traits.	[92]
Somatostatin	- Somatostatin receptor 2 located in NB tissue; - Olfactory NB cells express somatostatin receptor 2, which is a target for radionuclide imaging; - Radiopharmaceuticals (^177^Lu-octreotide, ^177^Lu-octreotate) were used for the treatment of human NB cell (CLB-BAR)-bearing mice; after treatment with these somatostatin analogs, pro-apoptotic and anti-apoptotic genes were regulated.	[107,108,109,110,113]
Substance P	- Neurokinin-1 receptors expressed in NB cells; - Promotes NB cell proliferation; - NB cell proliferation inhibited with neurokinin-1 receptor antagonsits (L-733,060, aprepitant).	[63,64,65,67]
Thyrotropin-Releasing Hormone	- Exerts anti-apoptotic effects in NB cells.	[70]
Vasoactive Intestinal Peptide (VIP)/Pituitary Adenylate Cyclase-Activating Polypeptide (PACAP)	- VIP level increased during NB differentiation; - VIP and PACAP promote NB cell differentiation into benign form by controlling vascular endothelial growth factor, vascular endothelial growth factor receptor, and hypoxia-inducible factor expressions; - NB cells express VPAC2 receptors; - PACAP, via PAC-1 receptors, promotes its own expression in NB cells; this is mediated by protein kinase A, extracellular signal-regulated kinase, and novel protein kinase isoforms, and also favors VIP/fos gene expressions; - PACAP-27 protects NB cells from apoptotic mechanisms, but VIP has no effect; - PACAP-38 did not promote NB cell proliferation, but the leukemia inhibitory factor favored such proliferation; - PACAP-38, via PAC-1 receptors, favors neuronal differentiation in NB cells.	[115,116,118,119,120,121,122,123]
Vasopressin	- Tolvaptan decreases NB cell proliferation and invasion and induces apoptosis; - Raloxifene reduces vasopressin mRNA levels; - Testosterone and estradiol, via estrogen receptors, control vasopressin expression.	[71,72,73]

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
