# Peer review of "Peptidergic Systems and Neuroblastoma"

_ijms, 2025, doi:10.3390/ijms26083464_

Round 1

Reviewer 1 Report

Comments and Suggestions for Authors

This review describes the involvement of neuropeptides in neuroblastoma progression. This is an emerging area of cancer research, with many promising results. The review does a job of collating all of the many studies in this area and also discusses future avenues of research. I think this review will be of value for a number of researchers in the cancer field. I have only minor suggestions for how to improve it.

  • Abstract: A bit too detailed, with many things not explained, like P2X7. Either make even longer, and explain things better, or preferably, simplify.

  • Introduction and/or start of main text: Define “peptidergic” a bit better. I assume from the peptides discussed in the review that they are mostly (only?) referring to neuropeptides (e.g., NPY). Or with “peptidergic” are they referring to any peptide secreted by a cell? Also, what is a peptide? Typically, proteins smaller than 50aa are denoted peptides, while larger proteins are denoted proteins.

  • I suggest describing a bit more clearly 1) the normal source of each peptide, 2) the normal function of each peptide, 3) whether the peptide is aberrantly produced by the neuroblastoma cells or acts upon them from its normal source. Maybe integrate this info into Figure 1 and/or Table1.

  • There is a lot of complexity regarding the nomenclature and origin of different neuropeptides regarding genes. For instance, PACAP27 and PACAP-38 are generated from the ADCYAP1 gene. And Substance P from the Tachykinin 3 gene. Basically, the ~50 defined neuropeptide genes generate ~150 cleavage products. Since the review mixes discussion of genetic insults driving neuroblastoma with peptidergic effects it would be good to clarify the gene/peptide connection a bit better throughout.

  • Neuropeptides act chiefly on GPCR receptors. But there are many of these, and there is some cross-talk such the some peptides act on overlapping GPCRs. I find the review somewhat vague regarding which GPCRs are being targeted by different drugs.

Author Response

REVIEWER 1

This review describes the involvement of neuropeptides in neuroblastoma progression. This is an emerging area of cancer research, with many promising results. The review does a job of collating all of the many studies in this area and also discusses future avenues of research. I think this review will be of value for a number of researchers in the cancer field. I have only minor suggestions for how to improve it.

Thank you very much for your comments, which help to understand/clarify the review much better. Corrections in the new version appear in red.

  1. Abstract: A bit too detailed, with many things not explained, like P2X7. Either make even longer, and explain things better, or preferably, simplify.

It has been shortened. See lines 9-26.

  1. Introduction and/or start of main text: Define “peptidergic” a bit better. I assume from the peptides discussed in the review that they are mostly (only?) referring to neuropeptides (e.g., NPY). Or with “peptidergic” are they referring to any peptide secreted by a cell? Also, what is a peptide? Typically, proteins smaller than 50aa are denoted peptides, while larger proteins are denoted proteins.

This has been explained in Introduction. Using the term “peptidergic”, we mean that the same peptide can be synthesized by nerve and non-nerve cells and for this reason we have avoided the term “neuropeptide”. See lines 44-47.

  1. I suggest describing a bit more clearly 1) the normal source of each peptide, 2) the normal function of each peptide, 3) whether the peptide is aberrantly produced by the neuroblastoma cells or acts upon them from its normal source. Maybe integrate this info into Figure 1 and/or Table1.

This has been clarified. See lines 47-56. We have chosen to explain the points suggested by the reviewer in the Introduction because the sources of peptides are diverse (nerve cells, tumor cells, endothelial cells, cells of the immune system) and the normal functions of peptides are numerous and varied as well as the pathological ones. Moreover, the activity of tumor cells is regulated by the tumor cell itself: it produces and releases peptides that bind to its own receptors exerting an autocrine action, and/or activates other tumor cells (paracrine action) and/or peptides are released into the blood (endocrine action) and also the activity of cancer cells is controlled by the release of peptides by other cells (nerve, endothelial or immune cells). Therefore, a double action mediated by the tumor cell itself and by others located in the tumor microenvironment can occur. Regarding the neuroblastoma, there are peptides known to be synthesized in tumor cells (ACTH, GRP, neuropeptide Y), but there are many others where this remains unknown. This demonstrates the degree of current ignorance regarding the role of peptides in neuroblastoma.  The functions of peptides can be different depending on where they are released and also depending on the concentration. Therefore, due to the great variability (sources and functions of peptides) that exists and the lack of knowledge in many aspects in neuroblastoma, we think it is more appropriate to write an explanatory text in the Introduction than to include the information in Figure 1 or Table 1. In addition, we think that this text will help the reader to understand it better.

  1. There is a lot of complexity regarding the nomenclature and origin of different neuropeptides regarding genes. For instance, PACAP27 and PACAP-38 are generated from the ADCYAP1 gene. And Substance P from the Tachykinin 3 gene. Basically, the ~50 defined neuropeptide genes generate ~150 cleavage products. Since the review mixes discussion of genetic insults driving neuroblastoma with peptidergic effects it would be good to clarify the gene/peptide connection a bit better throughout.

This important point has been mentioned. See lines 612-614.

  1. Neuropeptides act chiefly on GPCR receptors. But there are many of these, and there is some cross-talk such the some peptides act on overlapping GPCRs. I find the review somewhat vague regarding which GPCRs are being targeted by different drugs.

This very interesting point raised by the reviewer remains, like many others, to be investigated in neuroblastoma. In any case, the antagonists shown in Figure 3 are quite specific for the peptide receptors indicated and for this reason we have selected them. In this Figure, we have selected the most specific antagonists we found when performing the review. These peptide receptor antagonists counteract tumor cell growth and migration, and angiogenesis. This selection of antagonists is already a starting point for developing new research focused on neuroblastoma. That's why we have shown the chemical structures and that is also why it is important to know which peptide receptors the NB cells express as indicated in lines 679-683. See also lines 699-702.

Reviewer 2 Report

Comments and Suggestions for Authors

This is a well-written manuscript to summarize the peptidergic systems and neuroblastoma. The authors reviewed current evidences on peptides related researches including peptides, their receptors, agonists/antagonists from in vitro/in vivo or clinical trials. And they also highlight the gaps and pinpoint the future directions. As the authors stated "unfortunately much of the existing data regarding the above-mentioned aspects are fragmentary, scarce, or absent in many peptidergic systems and hence there is still a lot of basic research work to be done in NB". Researchers in this field would appreciate your excellent work and benefit from your paper. 

Author Response

REVIEWER 2

This is a well-written manuscript to summarize the peptidergic systems and neuroblastoma. The authors reviewed current evidences on peptides related researches including peptides, their receptors, agonists/antagonists from in vitro/in vivo or clinical trials. And they also highlight the gaps and pinpoint the future directions. As the authors stated "unfortunately much of the existing data regarding the above-mentioned aspects are fragmentary, scarce, or absent in many peptidergic systems and hence there is still a lot of basic research work to be done in NB". Researchers in this field would appreciate your excellent work and benefit from your paper. 

Thank you very much for your comment and the appreciation of the work for the scientific community interested in neuroblastoma. We think that better understanding the role of peptides in tumor development is a line of research that should be strengthened and, in some cases, re-opened due to the potential benefits it could provide in combating cancer. We deeply appreciate your comment.